

# Spatial Extent of New Particle Events over the Mediterranean basin from multiple ground-based and airborne measurements

Kevin Berland[1], Clémence Rose[1], Jorge Pey[2], Anais Culot[1], Evelyn Freney[1], Nikolaos Kalivitis[3], Giorgios Kouvarakis[3], José Carlos Cerro[4], Marc Mallet[5], Karine Sartelet[6], Matthias Beckmann[7], Thierry Bourriane[8], Greg Roberts[8], Nicolas Marchand[2] Nikolaos Mihalopoulos[3,9] and Karine Sellegri[1]

[1]Laboratoire de Météorologie Physique, CNRS UMR 6016, Université Blaise Pascal, Aubière, France

[2]Aix-Marseille Université, CNRS, LCE, UMR 7376, 13331, Marseille, France

[3]Environmental Chemical Processes Laboratory, University of Crete, Heraklion, Crete, 71003, Greece

[4]Laboratory of Environmental Analytical Chemistry, Illes Balears University, Palma, 07122, Spain

[5]Laboratoire d'Aérologie (LA), Université de Toulouse, CNRS, Toulouse, France

[6]CEREA, joint laboratory Ecole des Ponts ParisTech - EDF R&D, Université Paris-Est, 77455 Champs sur Marne, France.

[7]Laboratoire Interuniversitaire des Systèmes Atmosphériques (LISA), UMR-CNRS 7583, Université Paris-Est-Créteil (UPEC) et Université Paris Diderot (UPD), Institut Pierre Simon Laplace (IPSL), Créteil, France

[8]Centre National de Recherches Météorologiques, Météo-France, Toulouse, URA1357, France

[9]IERSD, National Observatory of Athens, P. Penteli, 15236, Athens, Greece

*Correspondence to*: Karine Sellegri (K.Sellegri@opgc.cnrs.fr)

**Abstract.** Over the last two decades, new particle formation (NPF), i.e. the formation of new particle clusters from gas-phase compounds followed by their growth to the 10-50 nm size range, has been extensively observed in the atmosphere at a given location, but their spatial extent rarely assessed. In this work, we use aerosol size distribution measurements performed simultaneously at Ersa (Corsica) and Finokalia (Crete) over a one-year period for analyzing the occurrence of NPF events in the Mediterranean area. The geographical location of these two sites, as well as the extended sampling period allow us to assess the spatial and temporal variability of atmospheric nucleation at a regional scale. Globally, Finokalia and Ersa show similar seasonalities in the monthly average nucleation frequencies, growth rates, and nucleation rates although the two stations are located more than 1000 km away from each other. Within this extended period, aerosol size distribution measurements were performed during an intensive campaign (July 3[rd] to August 12[th] 2013) from a ground based station on the island of Mallorca, as well as onboard the ATR-42 research aircraft. This unique combination of stationary and mobile measurements provides us with detailed insights into the horizontal and vertical development of the NPF process on the daily scale. During the intensive campaign, nucleation events occurred simultaneously both at Ersa and Mallorca over delimited time slots of several days, but different features were observed at Finokalia. The results highlight that the spatial extent of the NPF events over the Mediterranean Sea might be as large as several hundreds of kilometers, mainly determined by synoptic conditions. Airborne measurements gave additional information regarding the origin of the clusters detected above the sea. The selected cases depicted





contrasting situations, with clusters formed in the marine boundary layer or initially nucleated above the
continent or in the Free Troposphere (FT) and further transported above the sea.

**1 Introduction**
New particle formation (NPF) events have been widely observed in the atmosphere in different environments
(Kulmala et al., 2004) from remote areas at high altitude or latitude to polluted environments in different
climates (Cusack et al., 2013; Manninen et al., 2010; Pey et al., 2008; Yli-Juuti et al., 2011). However, the exact
mechanism and chemical species involved in the NPF process are not fully identified, especially regarding the
diversity of environments to consider. Thus most global climate models still do not represent well this process,
using parameterizations which are based upon a limited number of mechanisms and gaseous precursors, even
though they predict that it may contribute to a significant fraction of condensation nuclei (CN) and cloud
condensation nuclei (CCN) concentration at the global scale ( Makkonen et al., 2012; Merikanto et al., 2009;
Spracklen et al., 2008).
The different features of NPF events (frequency, intensity, duration etc…) may be influenced by meteorological
variables (temperature, relative humidity and solar radiation) (Birmili et al., 2003; Jeong et al., 2004; Sihto et al.,
2006; Young et al., 2007), but also by the availability of gaseous precursors, regarding both their nature and their
amount. It is thus necessary to describe the occurrence and characteristics of NPF over a large variety of
environments, and assess to what spatial extent these features can be applied to. Although the characteristics of
the NPF events have often been documented in the literature (Hirsikko et al., 2007; Manninen et al., 2010; Yli-
Juuti et al., 2009, 2011), analysis dedicated to their spatial extent are rare. This might be explained by the fact
that such studies require airborne measurements (Crumeyrolle et al., 2010; Rose et al., 2015) or multi-sites
datasets. Such datasets were analyzed by Hussein et al. (2009) who reported that NPF could take place in the
form of regional events over up to thousand kilometers in Scandinavia. Likewise, Dall'Osto et al. (2013)
observed regional NPF events occurring in the North-East of Spain. Using a similar methodology, Crippa and
Pryor, (2013) observed horizontal extents of a hundred kilometers for the NPF process in USA and Canada.
They also pointed out a significant variability of the NPF characteristics (formation and growth rates) within
these large scale events, suggesting that local signatures could superimpose to favorable synoptic conditions. In
order to allow for the analysis of the horizontal extent of NPF on a single station dataset, different methods based
on air mass back trajectory analysis and particle growth rates were also recently proposed (Kristensson et al.,
2014; Rose et al., 2015). The Nanomap tool developed by Kristensson et al., (2014) was reported to allow the
identification of nucleation areas up to 500 km far from the observation site. The main limitation of this last
method is due to the fact that the determination of the nucleation area directly depends on event characteristics
that sometimes cannot be accurately defined (i.e. the determination of the end of the nucleation process itself, or
the end of the growth process).
These few studies dedicated to the analysis of the horizontal extent of NPF were mainly conducted above
continental regions. Similar analysis in marine environments are crucially missing although they are of high
interest, as it was previously shown that in such pristine environments, cloud properties could be significantly
impacted by changes in the aerosol loading (Koren et al., 2014; Rosenfeld et al., 2014; Tao et al., 2012).
Although the Mediterranean area is particularly sensitive to the future evolution of atmospheric pollutants and





climate change, only a few studies relative to NPF in this area have been reported so far. Intensive campaigns
were conducted on the eastern Spanish coast, in Barcelona and at Montseny site (Cusack et al., 2013; Pey et al.,
2008), while long-term measurements are performed at the Finokalia (Crete) station (Kalivitis et al., 2008, 2012,
2015; Manninen et al., 2010; Pikridas et al., 2012). Frequencies of NPF were reported to range between 25% and
36%. The Mediterranean basin is at the cross section of many different influences: there is a strong
anthropogenic influence from densely populated coastal zones, which superimpose with marine and dust sources,
as well as with emissions from Mediterranean forests and shrublands that emit both terpenes and isoprene. This
geographical area is particularly exposed to high radiation compared to the rest of Europe, so that we expect a
strong contribution from photochemical processes.
In the framework of the projects CHARMEX-ADRIMED (Mallet et al., 2015) and CHARMEX-SafMed, a large
coordinated effort of the French community has been recently conducted to better characterize the physico-
chemical properties of the Mediterranean atmosphere. Measurements were conducted at ground-stations on
Mediterranean islands, such as Crete (Finokalia) and Corsica (Ersa) for an extended period of the years 2013-
2014 and Mallorca (Cap Es Pinar) for several weeks during 2013. Forty airborne research flights were also
performed during the summers 2013 and 2014. This vast dataset gave us a unique opportunity to characterize the
spatial extent of the NPF process in the Mediterranean basin. In this paper, we first report long term analysis of
NPF event characteristics measured at Ersa (from May 2012 to August 2013) and Finokalia (from January to
December 2013) using size distribution measurements, in order to assess the large scale space and time
variability of NPF. We then focus our study on the Special Operation Period (SOP) that took place during
summer 2013. During this SOP additional measurements took place in Mallorca (from July 3$^{rd}$ to August 12$^{th}$
2013) and aerosol particle size distributions and concentrations were measured onboard the ATR-42, which
allowed for a deeper analysis of the horizontal and vertical development of the NPF process at the daily scale.

**2  Experimental platforms, material and methods**
**2.1 Ground-based measurements**
Ground-based aerosol measurements reported in this work were performed at the Finokalia station (Crete) from
January to December 2013, at the Ersa station (Corsica) from May 2012 to August 2013, and at the Cap Es Pinar
station (Mallorca,) from July 3$^{rd}$ to August 12$^{th}$ 2013 (Figure 1). Within these measurements periods, some gaps
occurred in the Finokalia data set (from September 5$^{th}$ to October 15$^{th}$ 2013, 2013), due to participation of the
instrument in the ACTRIS (Aerosol Clouds and Trace gases Research Infrastructure) network mobility particle
size spectrometer workshop, and in the Ersa data set from September 1$^{st}$ to October 31$^{th}$ 2012), due to
instrumental failures.
The Finokalia station (35.24° N, 25.60° E) is located on the northern coast of Crete, Greece, at the top of a hill
(230 m a.s.l) facing the sea. There is no significant human activity within an area of approximately 15 km around
the station, mainly characterized by a scarce vegetation (Mihalopoulos et al., 1997). The closest large urban area
is the city of Heraklion, with 150 000 inhabitants, located 50 km West from Finokalia. Aerosols at the site are
mainly transported from the Southern-Eastern Europe and Northern Africa, and to a lesser extent from Central
and Western Europe (Kouvarakis et al., 2000; Sciare et al., 2008; Pikridas et al., 2010, 2012). At Finokalia,



aerosol particle size distributions were measured in the size range 8.8–848.7 nm, with a time resolution of 300 s,
with a custom-made Mobility Particle Size Spectrometer type (SMPS) (Wiedensohler et al., 2012). As
previously described by Kalivitis et al., (2015), the system operates with a closed-loop sheath air flow, with a 5:1
ratio between the sheath and the aerosol flow. It comprises a Kr-85 aerosol neutralizer (TSI 3077), a Hauke
medium differential mobility analyzer (DMA) and a TSI-3772 condensation particle counter (CPC). The system
is operated following the recommendations of Wiedensohler et al., (2012), thus meeting the European
infrastructure ACTRIS project requirements for quality insurance.
The Ersa station is located on the northern tip of Corsica Island, on Cape Corsica (43.00° N, 9.30° E, 530 m
a.s.l.). On this part of the island the wind can be very strong with frequent windstorms (78 days in 2007 with a
wind speed stronger than 28 m s$^{-1}$). Climate in Corsica is characterized by moist winters and dry summers, with
less than 100 rainy days per year (Lambert et al., 2009). Aerosols reaching the site are of variable types,
including mineral dust particles from North Africa, anthropogenic and biomass burning aerosols mainly
originating from densely populated coastal areas located in Eastern Spain, France and Italy, and marine aerosols,
both from the Mediterranean Sea itself but also from the Atlantic Ocean (Mallet et al., 2016; Nabat et al., 2013).
The Cape Corsica peninsula is a remote site, excluding important local anthropogenic sources that could affect
the in-situ measurements, and surrounded by a scarce Mediterranean vegetation (Mallet et al., 2016). At Ersa,
aerosol size distributions were measured with a scanning mobility particle sizer (SMPS TSI 3080, associated to a
CPC TSI 3010) in the size range 10.9-495.8 nm with a time resolution of 300 s.
The Cap Es Pinar station is located on the Northeastern side of the Mallorca Island (39.88° N, 3.19° E, 20 m
a.s.l.), on a peninsula between the Alcudia and Pollença bays. The station was placed in one of the buildings
belonging to the Spanish Ministry of Defense in his Cap Es Pinar facilities. The area is densely forested by
Mediterranean shrublands and pine trees and the access to the facilities is restricted. Urban centers, the Alcudia
and Pollença harbors and main roads are located at least 10 Km from the site. Particle size distributions were
measured in the size range 15-600 nm with a time resolution of 300 s using a TSI SMPS, with a 3081 long DMA
and a 3776 CPC.
**2.2 Airborne measurements**
Airborne measurements were carried out on board of the ATR-42, French research aircraft operated by SAFIRE
(Service des Avions Français Instrumentés pour la Recherche en Environnement). Figure 1 shows also the
aircraft trajectory during the flights performed on July 30$^{th}$ and August 1$^{st}$ which are investigated in the next
sections of the present work. The aerosol size distribution in the 20-485 nm diameter range was measured with a
time resolution of 130 s using the SMPS system previously described in (Crumeyrolle et al., 2010) which
includes a CPC (TSI, 3010), a differential mobility analyser (DMA) and a krypton aerosol neutralizer. The total
concentrations of aerosols larger than 10 nm ($N_{10}$) and larger than 3 nm ($N_3$) were measured using a custom-
made CPC dedicated to aircraft measurements (Weigel et al., 2009) and a CPC TSI 3025 type, respectively. The
concentration of particles in the size range 3 - 10nm ($N_{3-10}$) was calculated as the difference between $N_3$ and $N_{10}$.
After analysis of the variability of $N_{3-10}$ apart from nucleation periods, we found that $N_{3-10}$ concentrations are
above the variability of the two CPC concentration difference, corresponding to a threshold of 395 cm$^{-3}$. For
more details on the airborne instrumentation and data analysis procedure, the reader is referred to Rose et al.,

152 (2015).






## 3 Data analysis

### 3.1 NPF events classification

From ground-based observations, NPF were classified according to Dal Maso et al (2005) into four categories: events days, including classes I and II, undefined and non-events days. Class I events are characterized by a strong increase of sub-25nm particles concentrations, their persistence over a period of more than an hour and a clear growth of the nucleation mode particles towards larger sizes during the following hours. Class II events have the same characteristics as Class I events, except that they may be less intense or show a discontinuity in the growth of the clusters. Days are considered undefined when the newly observed particles are detected only from the Aitken size and/or when they do not grow during the course of the day.

### 3.2 Particle formation and growth rates calculations

Particle formation and growth rates are key entities to assess the strength of events belonging to Class I and II. Growth rates (GR) were calculated from the SMPS nucleation mode concentrations (16.3-20.2 nm) using the "maxima" method from Hirsikko et al.(2005). The time corresponding to the maximum concentration was first determined for each of the SMPS size channels in the range $16.3 - 20.2$ nm by fitting a normal distribution to the concentrations. The growth rate was then derived from a linear least square fit through these time values.

From this growth rate, we derived the total particle formation rate at 16 nm ($J_{16}$), similarly as in Dal Maso et al. (2005) using the following equation (Eq.1) :

$$J_{16} = \frac{dN_{16}}{dt} + CoagS_{16} * N_{16} + \frac{GR_{16.3-20.2}}{(20.2 - 16.3)nm} * N_{16} \qquad (1)$$

$CoagS_{16}$ is the coagulation sink of 16 nm particles on larger particles, $N_{16}$ is the total concentration of 16.3-20.2 nm particles and $GR_{16.3-20.2}$ is the growth rate corresponding to the same diameter range.

## 4 Results and discussion

### 4.1 Yearly statistical analysis of NPF events characteristics at two ground-based stations

The goal of this first section is to provide an overview of the seasonal variability of NPF in the Mediterranean area, and some insights into the spatial homogeneity of the NPF occurrence over the basin.

### 4.1.1 NPF Events frequency and types

The yearly average NPF frequencies, calculated as the number of event days over the total number of measurement days, are very similar at the two stations, being 36% and 35% at Finokalia and Ersa, respectively. A comparable value is reported by Pikridas et al. (2012) at Finokalia, with a yearly average frequency around



185 33% calculated over a year from April 2008 to April 2009. At both stations, the NPF frequency shows a clear

186 annual cycle with the highest frequencies observed during spring (52% in May for Finokalia and 56% in April

187 for Ersa), and the lowest in autumn (Fig. 2). A similar seasonal variation was already reported for the Finokalia

188 station, with a slight time offset of the NPF frequency peak, which was found in February-March (Pikridas et al.,

189 2012). Higher NPF frequencies are frequently observed during spring (April-May-June) compared to the rest of

190 the year at European stations. They are mainly explained by the onset of biogenic emissions and increased solar

191 radiation (Manninen et al., 2010). The classification of the measurement days into the different categories (Fig.

192 3) shows that the occurrence of type I events in Finokalia follows the same seasonal variation as the total NPF

193 frequency, being maximum during the spring season (up to 26% of all days). This indicates that the spring

194 season is favorable to both formation of new particles and their growth to larger sizes. Type II events are

195 globally the most frequent, representing between 13% and 31% of all measurement days with no clear seasonal

196 variation. In contrast, undefined days are not frequently observed in Finokalia, around 9% on average. Very

197 similar features are observed in Ersa: type I events show the highest frequency of occurrence during spring and

198 summer (up to 32% of all days in August), while they represent less than 10% of the measurement days during

199 winter. The frequency of occurrence of type II events is on average 19%, with no clear seasonal variation.

### 4.1.2 Growth rates and particle formation rates

201 Particle formation and growth rates were calculated for type I events in order to characterize the strength of

202 the events observed at the different stations. The yearly median particle growth rates in the range 16 – 20 nm

203 ($GR_{16-20}$) are 7.10 and 16.7 nm h$^{-1}$ at Ersa and Finokalia, respectively (Table 1). The values obtained at Finokalia

204 are on average slightly higher compared to those reported by Manninen et al. (2010) in the range 7 – 20 nm (1.8

205 – 20 nm h$^{-1}$, mean value 4.4 nm h$^{-1}$). More generally, the values calculated in this work are on average slightly

206 higher compared to those obtained at other European coastal sites such as Cabauw (2.1 - 19 nm h$^{-1}$, mean value

207 6.7 nm h$^{-1}$) and Mace Head (2.7 – 10 nm h$^{-1}$, mean year value 5.4 nm h$^{-1}$) (Manninen et al., 2010). Figure 4

208 displays the annual variation of the particle growth rates (GR) at Ersa and Finokalia. At Ersa, the GR have the

209 same seasonal variation as the NPF frequency, with higher values in spring compared to the rest of the year. At

210 Finokalia, the GR seasonality is not as clear as in Ersa. However the seasonality in Finokalia is rather biased

211 because there are only few class I events during summer.

212 The yearly median particle formation rates are 0.16 cm$^{-3}$s$^{-1}$ in Ersa and 0.26 cm$^{-3}$s$^{-1}$ in Finokalia (Table1). These

213 values are slightly lower than the $J_{10}$ values reported by Kulmala et al. (2004) from several coastal sites and ship

214 campaigns conducted in the Baltic, Atlantic and Pacific areas (0.4 – 1.5 cm$^{-3}$s$^{-1}$). The values calculated in this

215 work are, to our knowledge, the first reported for the formation of nucleation mode particles (10 – 20 nm) in the

216 Mediterranean basin. As reported on Fig. 5, the median $J_{16}$ particle formation rates also follows a seasonal

217 variation similar to the NPF frequency at both stations, with higher values in spring (March with 0.56 cm$^{-3}$s$^{-1}$ for

218 Finokalia, and April with 0.66 cm$^{-3}$s$^{-1}$ for Ersa). In contrast, lower $J_{16}$ are observed in early winter and mid-

219 summer at both stations. Available precursors for initiating nucleation in spring, presumably due to the onset of

220 biogenic emissions and increased solar radiation, are likely of the same origin than thus necessary for growing

221 clusters up to 16 nm.

222 It is worth noticing that in Ersa, even though NPF frequencies are weaker in autumn compared to spring,

223 particle formation rates are comparable. This last observation suggests that, despite being less frequent, favorable



conditions for NPF can be found during autumn and lead to events with the same intensity as in spring, when
radiation and biogenic emissions are on average higher compared to the rest of the year (Manninen et al., 2010).
Factors explaining the seasonal variation of nucleation frequency, nucleation rates and growth rates can be the
availability of condensable gases. The amount of such precursors results from a balance between a combination
of emissions and radiation that favor their production, and their loss onto preexisting particles. In order to assess
the influence of the preexisting aerosol population on NPF, we calculated the condensational sink (Cs) according
to Pirjola et al. (1999). The Cs was first derived from SMPS measurements for the whole measurement period at
both stations and was finally averaged over the two-hour period prior to the onset of NPF events. On non-event
days, the Cs was averaged over the two-hours time period prior to the mean time at which the NPF is triggered
on event days, i.e. around 11:00 (UTC) in Finokalia and 12:00 (UTC) in Ersa. The annual variation of the
median Cs derived from these averaged values is reported for event and non-event days on Fig. 6.
The Cs has a strong seasonal cycle with a clear maximum during summer at both stations. This observation may
explain the lower NPF frequencies, formation rates and growth rates that are on average observed during this
season, that otherwise shows high radiation and high biogenic emissions. In addition, at both stations, the Cs is
on average higher during non-event days. This confirms that the Cs is likely a limiting factor for the occurrence
of a NPF event at these stations. This was already pointed out by Hamed et al. (2010), Kulmala et al. (2005) and
Manninen et al. (2010) for the European atmosphere covering both the industrialized areas and boreal forest
environment respectively. The Cs is on average higher in Finokalia, especially during spring and summer with
monthly Cs twice as high compared to Ersa. It is worth noticing that large particles up to 848 nm are accounted
for in the Cs calculation in Finokalia, while the upper size limit is 495 nm in Ersa. However, particles above 500
nm only have a weak impact on the Cs values due to their low concentration, and thus do not explain the
differences which are seen between the sites. At Finokalia, N/NE winds dominate during summer, bringing high
concentrations of anthropogenic aerosol that have aged when passing over the sea before reaching the station,
thus leading to high Cs values. The fact that NPF frequencies, nucleation rates and growth rates are comparable
at the two stations indicates that the sources of condensable gases are likely to be significantly higher in
Finokalia compared to Ersa in order to compensate for the large condensational sink measured at the Crete
station.
Globally, Finokalia and Ersa show similar seasonality in the average nucleation frequency, growth rates and
nucleation rates although the two stations are more than 1000 km away from each other. It is worth mentioning
that during the period of interest, 104 event days were observed at Finokalia and 96 at Ersa, among which 31
(with 8 events of class I) occurred at both stations at the same time These results could indicate that the spatial
extent of NPF events over the Mediterranean basin are of the order of magnitude of the distance between the two
stations. However, we will downscale the comparison of occurrence and characteristics of events at the daily
resolution (rather than monthly), in order to further investigate this hypothesis.
**4.2 Intensive campaign during summer 2013**
**4.2.1 Ground-based measurements - overview**





In this section, we focus on the Special Observation Period (SOP) that took place from June 3$^{rd}$ to August 12$^{th}$ in
the frame of the CHARMEX project. During this period, number size distribution measurements were
additionally conducted at the Mallorca station (Cap Es Pinar).
Figure 7 shows the SMPS particle size distributions recorded at the three ground-based stations during the SOP.
We clearly observe similar trends in the evolution of the particle size distributions in Ersa and Cap Es Pinar, with
three distinct NPF periods during which NPF events occurred daily over several days (First period from July 4$^{th}$
to July 9$^{th}$, second period from July 28$^{th}$ to August 3$^{rd}$ and third period from August 9$^{th}$ to August 12$^{th}$) (see Table
S1). This observation would confirm the spatial extent of NPF events at a large scale. However, these periods of
intense NPF activity are not observed in Finokalia, where both the occurrence and strength of NPF events seem
to be more homogeneous over the SOP. These contrasting observations might be explained by an environmental
contrast between the eastern and western part of the Mediterranean basin.
In order to further investigate the link that might exist between the events observed at the three stations, we first
chose to focus our analysis on three specific days that belong to the three different NPF periods identified: July
5$^{th}$, July 29$^{th}$ and August 9$^{th}$ are presented as case studies.
**4.2.2 Ground-based measurements: Case studies**
We calculated the total formation rate of 20 nm particles ($J_{20}$) using particle growth rates $GR_{15-25}$ (Tab. 2) for the
three cases: July 5$^{th}$, July 29$^{th}$ and August 9$^{th}$. We first shortly describe the NPF events observed on the 5$^{th}$ and
29$^{th}$ of July (fully described in the supplementary) and then illustrate in more details the events observed on the
9$^{th}$ of august that have the most similarities between sites.
On July 5$^{th}$, although NPF occurred both at Ersa and Cap Es Pinar,  the time evolution of particle concentrations
are very different from one site to the other. Particles of the smallest size range are detected in the morning at
Ersa, but ony later in the afternoon at Cap es Pinar, and at larger sizes and lower concentrations (Fig S1). The
24-hour  air mass back trajectory analysis shows that air masses arriving at both stations are of northerly origin
(Fig S2). Hence it is unlikely that particles formed during the NPF event detected at Ersa in the morning have
been transported west and detected later in the afternoon at Cap Es Pinar. The calculation of the nucleation area
based on the NPF duration, growth rate and wind speed (see suppl. material), leads to a relatively small area of 9
km (Ersa) to 40 km (Cap Es Pinar), that does not allow further conclusions on the simultaneity of a large NPF
covering the spatial area of both stations. The event of July 29$^{th}$ was detected from the lowest sizes of the SMPS
at both stations with the same intensity (similar $N_{15-20}$ and $J_{20}$), and show similar features (Fig. S3), but was
detected one hour earlier at Cap Es Pinar than at Ersa. Air masses were from the northern sector at Cap Es Pinar,
and then turned west towards Ersa (Fig. S5).
In Finokalia, both for July 5$^{th}$ and July 29$^{th}$, significant $N_{15-20}$ concentration are also detected during the
nucleation hours, but in the form of a succession of peaks that do not show the usual feature of a clear NPF event
(with a continuous growth).
On August 9$^{th}$, newly formed particles are detected in air masses originating from the near Southern area in Ersa
and from Northwestern sector in Cap Es Pinar (Fig.9). The concentration of  particles mesured in the first SMPS
size channels in Ersa (11-15 nm) does not present very marked variations, while $N_{15-20}$ displays more significant





changes in the course of the day. These observations might suggest that unlike previous events, NPF may not be
initiated at the station itself, but rather in a neighbouring area (Fig. 8). Similar features are observed at Cap Es
Pinar, with significant variations of the particle concentration in the size range 15-20 nm, as on July 29th. The
temporal evolutions of $N_{15-20}$ and $N_{20-25}$ have similar structures at both stations between 10:00 and 16:00 UTC,
suggesting that NPF could occur simultaneously at both sites. Additional peaks of $N_{15-20}$ and $N_{20-25}$ are detected
earlier in the morning at Cap Es Pinar (7:20 and 9:00 UTC), while they are not detected in Ersa. Beside the
simultaneity of the process, NPF events detected at the two sites also display very similar characteristics, both
regarding particle growth (4.3 and 3.8 nm h$^{-1}$, for Ersa and Cap Es Pinar, respectively) and formation rates (4.83
and 4.17 cm$^{-3}$ s$^{-1}$, for Ersa and Cap Es Pinar, respectively). Instrumental failure did not allow similar analysis at
Finokalia. Figure 9 shows the estimate of the nucleation areas for the two stations. Concerning Cap Es Pinar, the
place where nucleation initially occurred is at least 49 km upstream the station.
Since all air mass back trajectories computed during the time period of interest are very local (at least during the
24 hours before their arrival at the site), we may hypothesis that NPF is occuring over the whole area close to
Mallorca where air mass backtrajectories overlap. Concerning Ersa, the nucleation of 20 nm particles latter
observed at the site is at least initiated 45 km upstream the station. The three case studies showed that NPF
events could be detected, with some time offset, on two remote stations separated by several hundred kilometers
in the Mediterranean area. In particular for the case of August 9th, the fact that these events can be detected in air
masses from different origins suggest that the NPF is, for both sites, initiated above the sea, either in the marine
boundary layer or higher in the free troposphere. In any case, the NPF process is likely not subject to the
availablility of precursors that would be specific to the air mass type reaching the sites. It could rather depend on
synoptic meteorological conditions at the European scale, including low condensational sinks following
precipitations periods. Indeed, the analysis of the meteorological conditons along backtrajectories shows that
precipitation did occur prior to their arrival at both stations on   July 29th (during the passage of low pressure
systems), but not on the two other case studies. The minimum areas that we determined for nucleation onset at
both sites do not overlap. However, the estimates we obtained are some lower limits of the actual values, and
there are no elements which could justify that the NPF is interrupted between both sites. Airborne measurements
will be used in the next section to further invastigate this aspect. In addition, these flights will allow an analysis
regarding the origin of the clusters and their precursors, from the marine boundary layer or from the upper levels
of the atmosphere, as previously shown by Rose et al. (2015a).
**4.2.3 Airborne measurements**
Among the 11 flights performed during the period, particles in the lowest size range ($N_{3-10}$) were not observed
during 7 of the flights, in agreement with no NPF events detected at the Ersa and Cap Es Pinar stations.  Two
flights detected elevated concentrations of $N_{3-10}$ and $N_{10-20}$ in agreement with NPF events at Ersa.
The first event to be investigated was observed on July 30th. Regarding aircraft measurements, the analysis was
focused on the flight legs performed at constant altitude and during which $N_{3-10}$ concentrations were above the
threshold value (Fig 10a). The first part of the flight was performed at low altitude (~ 215 m a.s.l.) from the
french coast towards the Ersa station and at higher altitudes (~ 3400 m a.s.l.) during the second part of the flight
from the Ersa station towards the coast. Based on Fig. 10, small particles ($N_{3-10}$) were detected at both altitudes
and over a large area included in a 219 × 131 km rectangle. On the low altitude flight section, $N_{3-10}$ are



decreasing from the Northeastern part of the flight track to the Southwestern one. This would indicate a source of
nanoparticles originating from the continent and progressively diluted in the marine boundary layer. However,
despite a high variability, $N_{3-10}$ were on average higher at high altitude, with average concentrations of
$3805\pm1555$ cm$^{-3}$ compared to $2040\pm2174$ cm$^{-3}$ at lower altitude. This last observation supports the results of
Rose et al. (2015a) who reported that nucleation could be enhanced at high altitude above the Mediterranean Sea
and connected to different sources at low altitude.
In order to explore the link that may exist between the events detected simultaneously from the aircraft and from
the ground, we first investigated the origin of the air masses. Figure 10b shows the 72 hour back trajectories of
the air masses sampled by the ATR-42 every 10 min along the flight path as well as the 72 hour back trajectories
of the air masses that reached Ersa in the meanwhile at 13:00, 14:00 and 15:00 UTC. During the first part of the
flight performed at low altitudes, the aircraft flew in Southern air masses which all passed over the continent
before sampling and became more local as the aircraft approached Ersa.  In contrast, the air masses sampled at
high altitude are from Western origin, so that they also passed over the continent, but did not display any local
features.
In addition, Fig. 11 shows the evolution of the particle size distributions measured onboard the ATR-42 and at
Ersa. The spectra are color coded according to the position of the aircraft indicated in the insert included in the
middle panel of Fig. 11. At Ersa, the shape of the particle size distribution remains similar during the whole
measurement period, with a nucleation mode around 20 – 25 nm, an Aitken mode around 50 – 60 nm which
clearly dominates the spectra and two accumulation modes, respectively around 110 and 220 nm. These modes
were identified when fitting the SMPS size distributions with four Gaussian modes using the methodology
described in Rose et al. (2015a). In contrast, the size distributions provided by the SMPS onboard the ATR-42
show significant variations. Lower concentrations are on average observed at higher altitude for the whole
diameter range but with more significant changes of the nucleation and Aitken modes. The shape of the size
distribution is also impacted by the location of the plane, especially at low altitude. In fact, the total particle
concentration decreases as the aircraft moves further off the southern coast of France, with, again, a more visible
impact on nucleation and Aitken modes.
These last observations, together with the air mass back trajectory analysis shown on Fig. 10.b, suggest that for
this first event, new particles were initially formed at low altitude over the continent and further transported
above the sea to be finally detected over a large area, and more especially in Ersa. Decreasing particle
concentrations observed while moving further off the continent make less probable the hypothesis of new small
particles formation from an additional marine source, but rather depict the effect of dispersion process that may
have taken place during particle transport.
The second event included in this analysis was observed on August 1$^{st}$. Compared to the previous study case, the
flight was performed over a larger area ($172 \times 247$ km rectangle) located further away west from Ersa and at a
relatively low constant altitude ($\sim 500$ m a.s.l.). $N_{3-10}$ concentrations above the threshold value were detected
along the flight path (Fig. 12) and compared well, on average, with the concentrations obtained at low altitude
during the flight performed on July 30$^{th}$ ($2483\pm2767$ cm$^{-3}$). However, $N_{3-10}$ concentrations occurred as bursts,
with no clear spatial gradient as previously reported for flight performed on July 30th. The analysis of air mass
back trajectories is shown on Fig 12.b. North-Eastern air masses were sampled at the beginning and at the end of





the flight, with northern air masses in between. Air masses from the North were also detected at Ersa and it is
worth noticing that, at least during the first part of the flight, the air masses that reached the aircraft had all
passed over Ersa region.
The evolution of the particle size distributions together with the location of the aircraft is shown in Fig. 13.
Unlike during the flight performed on July 30$^{th}$, the shape of the distributions measured onboard the ATR-42
remains similar during the whole measurement period despite the changing origin of air masses. In contrast, the
shape of the particle size distributions measured at Ersa shows a significant variability. Especially, the nucleation
mode displays increasing diameters from 20 to 30 nm and highly variable concentrations. Also, total
concentrations from Ersa are significantly higher compared to those measured onboard the ATR-42.
In order to further investigate the origin of the nucleation mode particles and the connection that may exist
between ground based and airborne measurements, we compared the diameters of the corresponding nucleation
modes. For that purpose, Fig. 14 shows the ratio of the nucleation mode diameter obtained onboard the ATR-42
over that from Ersa as a function of the distance between the aircraft and the station. This ratio is in the range 0.6
– 1.2, with on average decreasing values while increasing the distance between the two measurement points.
Nucleation mode diameter getting smaller along the air mass back trajectory above the sea could be the result of
intense inputs of nucleated particles initially below the SMPS size detection limit and feeding the nucleation
mode as they grow, as confirmed by the occurrence of $N_{3-10}$ nm particles detected in the ATR-42.  In this
particular case, particles detected in the nucleation mode observed onboard the ATR-42 would be the result of an
event occurring above the sea from marine precursors, which superimposes with a preexisting particle mode.

## 5 Conclusion

We investigated the occurrence of NPF in the Mediterranean area using particle size distributions measured at
three ground-based stations (Ersa, Cap Es Pinar and Finokalia) as well as airborne measurements performed in
2012 – 2013 in the frame of the CHARMEX-ADRIMED and CHARMEX-SafMed projects.
The analysis of long-term datasets from Ersa and Finokalia first revealed similar features, although the two
stations are more than 1000 km away from each other. Especially, almost equal annual NPF frequencies were
reported (36% and 35%, for Finokalia and Ersa, respectively) and similar seasonal variations of both the NPF
frequency and characteristics, i.e. particle formation and growth rates, were observed. The NPF process was on
average favored during spring, both in terms of occurrence and intensity, most probably because of increased
amounts of precursors from biogenic origin and higher solar radiation, thus allowing for more efficient
photochemistry processes.
This investigation, initially performed at a monthly resolution was downscaled in a second step at the daily
resolution over a two months period, in order to further better assess the simultaneity of NPF over a large part of
the Mediterranean basin. Three nucleation periods of several days appeared to clearly occur simultaneously at
Ersa and Cap Es Pinar, and less clearly at Finokalia. Three case study events were selected within these three
distinct NPF periods for a more detailed analysis. These three case studies showed that NPF events could be
detected, with some time offset, on two remote stations separated by several hundred kilometers in the





Mediterranean basin, without the stations being directly linked to eachother within a single air mass trajectory.
While featuring local characteristics, the occurrence of NPF events was not likely dependant on the
availablility of precursors that would be specific to the air mass type reaching the sites, but rather on synoptic
meteorological conditions at the European scale.
Airborne measurements were finally used to further investigate the horizontal and vertical extent of NPF, and to
determine the origin of the clusters and their precursors. Two case studies were again selected within the NPF
periods identified previously from ground-based observations,during which newly formed clusters were
observed onboard the ATR-42 and from Ersa on the same day. The selected events depicted contrasting
situations where particles were initially probably formed above the continent for one of them, both in the
boundary layer and in the free troposphere, and probably formed above the sea for the other. In each case,
clusters were detected over large areas confirming that NPF may be dependent on large scale processes.
This work, together with the previous study by Rose et al. 2015, demonstrates the occurrence of NPF in the
Mediterranean basin, thus highlighting the possibility for the process to be triggered above open seas. Those
results are of great interest to improve the parameterizations of nucleation in models, which actually only
consider a limited number of precursors, commonly including sulfuric acid and ammonia but excluding those
more specifically emitted in the marine atmosphere. Model predictions would also benefit from the analysis of
the vertical extent of the NPF process provided in these studies. Besides the identification of preferential
altitudes for the occurrence of the process, these results aid understanding the transport of the newly formed
clusters and their precursors between the boundary layer and the free troposphere. Future studies should focus on
understanding the chemical precursors that contribute to these new particle formation processes.
**Ackowledgements**
This study was performed with the financial support of the French National Research Agency (ANR) project
ADRIMED (contract ANR-11-BS56-0006) the ANR project SAf-Med "" (Grant Number: SIMI-5-6 022 04) and
is part of the ChArMEx project supported by ADEME, CEA, CNRS-INSU and Météo-France through the
multidisciplinary programme MISTRALS (Mediterranean Integrated Studies aT Regional And Local
Scales). The Financial support for the ACTRIS Research Infrastructure Project by the European Union's Horizon
2020 research and innovation program under grant agreement no. 654169 and previously Under grant agreement
no. 262254 in the 7th Framework Programme (FP7/2007–2013) is gratefully acknowledged.

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







**Tables**
**Table.1 Annual median formation rate, annual growth rate and annual Cs, with percentiles in Ersa and Finokalia.**

| | $J_{16}$ $(cm^{-3}s^{-1})$ | | | $GR_{16-20}(nm.h^{-1})$ | | | $Cs$ $(s^{-1})$ | | |
|---|---|---|---|---|---|---|---|---|---|
| | 25th perc. | med. | 75th perc. | 25th perc. | med. | 75th perc. | 25th perc. | med. | 75th perc. |
| Ersa | $1.4\times10^{-1}$ | $1.6\times10^{-1}$ | $3.0\times10^{-1}$ | 6.6 | 7.1 | 12.2 | $3.3\times10^{-3}$ | $4.1\times10^{-3}$ | $4.6\times10^{-3}$ |
| Finokalia | $1.9\times10^{-1}$ | $2.6\times10^{-1}$ | $2.8\times10^{-1}$ | 10.4 | 16.7 | 25.6 | $3.4\times10^{-3}$ | $6.2\times10^{-3}$ | $9.3\times10^{-3}$ |


**Table 2: Average growth rates and formation rates computed for individual events at Ersa and Mallorca**

| | Ersa | | Mallorca | |
|---|---|---|---|---|
| | $GR_{16-20}nm.h^{-1}$ | $J_{16}$ $cm^{-3}.s^{-1}$ | $GR_{16-20}$ $nm.h^{-1}$ | $J_{16}$ $cm^{-3}.s^{-1}$ |
| July 5th | 16.4 | $2.44\times10^{-1}$ | 7.8 | $0.41\times10^{-1}$ |
| July 29th | 8.9 | $7.88\times10^{-2}$ | 4.8 | $7.75\times10^{-2}$ |
| August 9th | 4.3 | $4.83\times10^{-2}$ | 3.8 | $4.17\times10^{-2}$ |





**Figures**

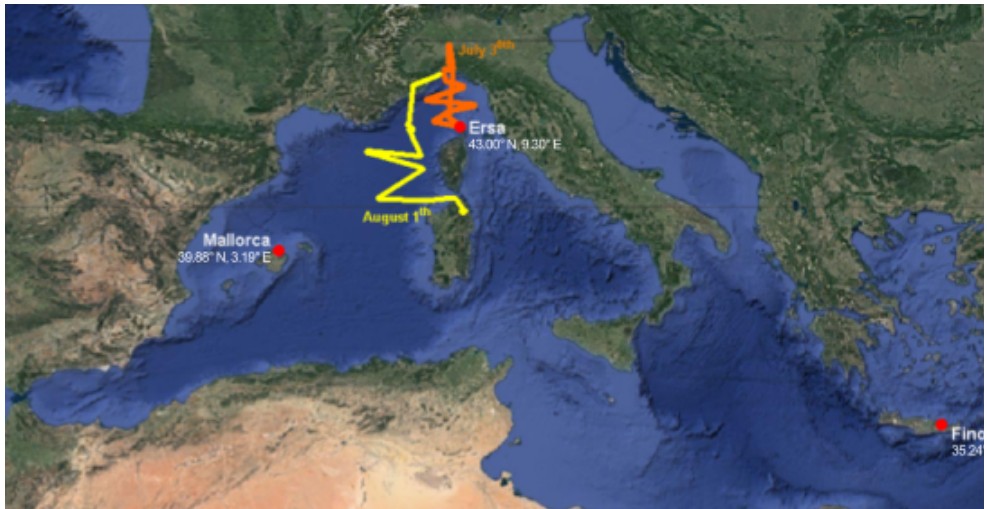


**Figure 1: Localization of stations Ersa (Corsica), Finokalia (Crete) and Cap Es Pinar (Mallorca) and of the aircraft**
**flight paths on July 30th and August 1st.**

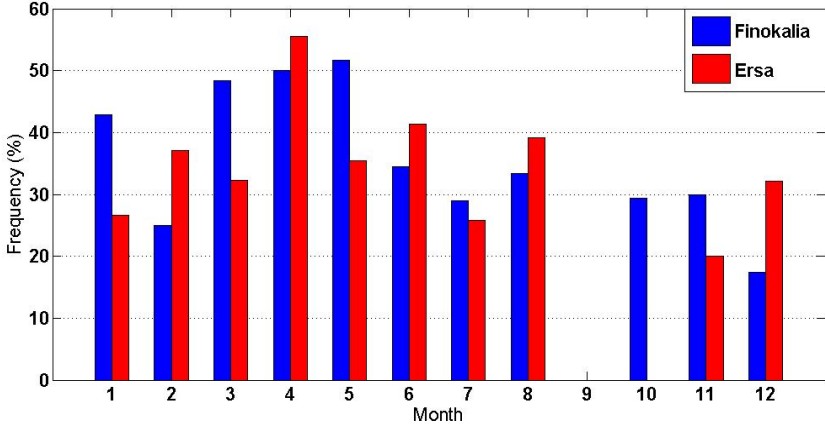


**Figure 2: Monthly mean (January to December) NPF frequencies at Finokalia and Ersa.**




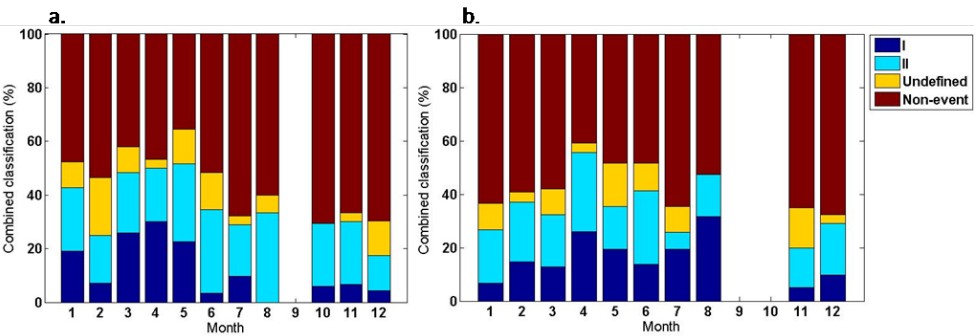


**Figure 3: Monthly classification of the days into event (I and II), undefined and non-event categories in Finokalia (a)**
**and Ersa (b)**

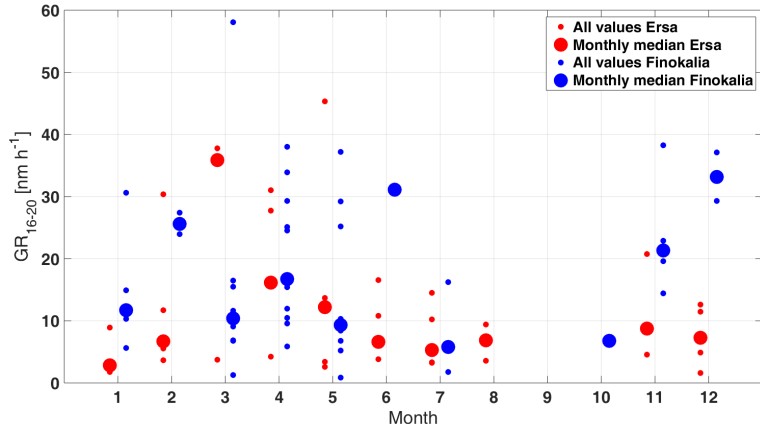



**Figure 4: Annual variation of particle growth rate calculated for the range 16 – 20 nm at Ersa (May 2012 – August**
**2013) and Finokalia (January – December 2013) for type I events. Small dots represent all values while large dots**
**stand for median values.**






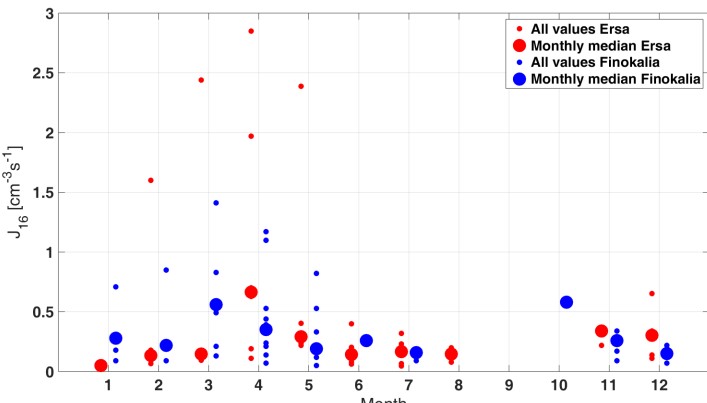


**Figure 5: Annual variation of the 16 nm particle formation rate calculated for the range 16 – 20 nm at Ersa (May**
**2012 – August 2013) and Finokalia (January – December 2013) for type I events. Small dots represent all values while**
**large dots stand for median values.**







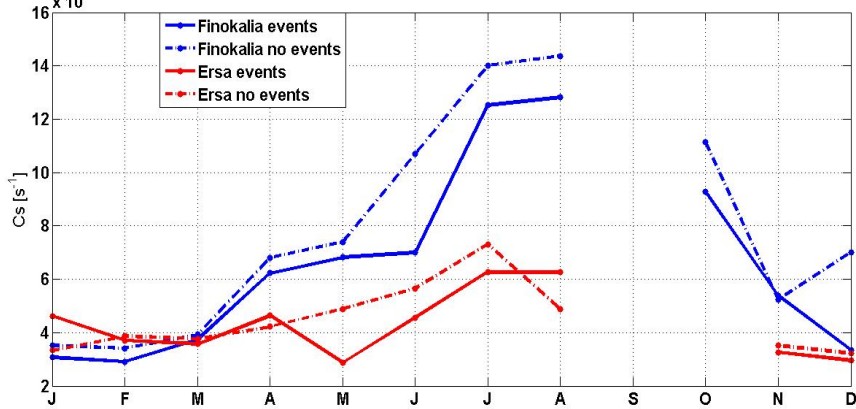


**Figure 6: Median values of condensation sink (Cs) during event and non-event days in Finokalia and Ersa.**



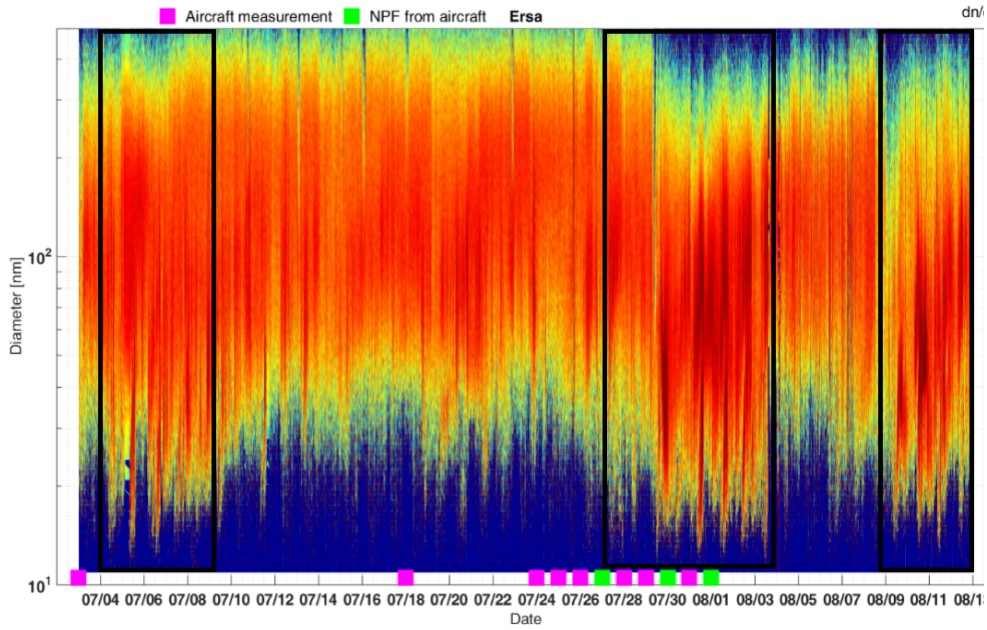


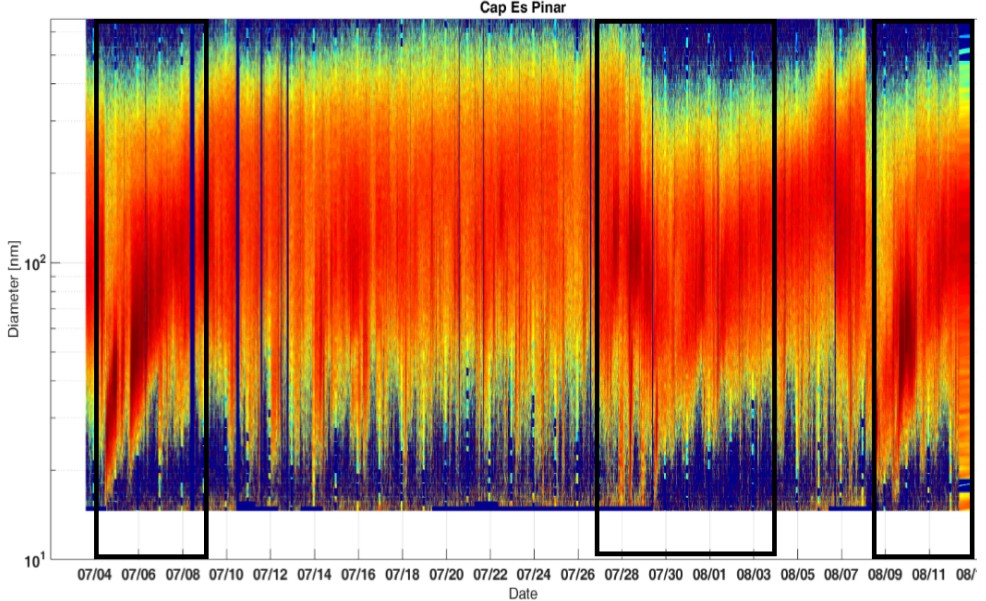




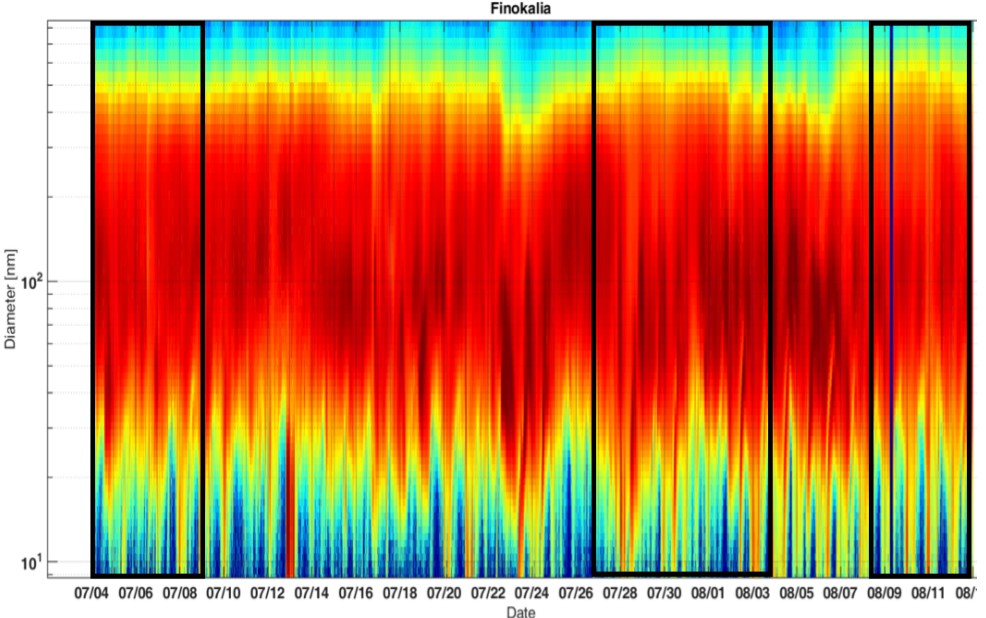


**Figure 7: SMPS particle number size distribution in Finokalia, Ersa and Cap Es Pinar (Mallorca) during the intensive campaign. The three NPF episodes observed at large scale are marked on the spectra in the black boxes. The days of occurrence of the ATR-42 flights are also shown, together with the detection of NPF from these airborne measurements.**

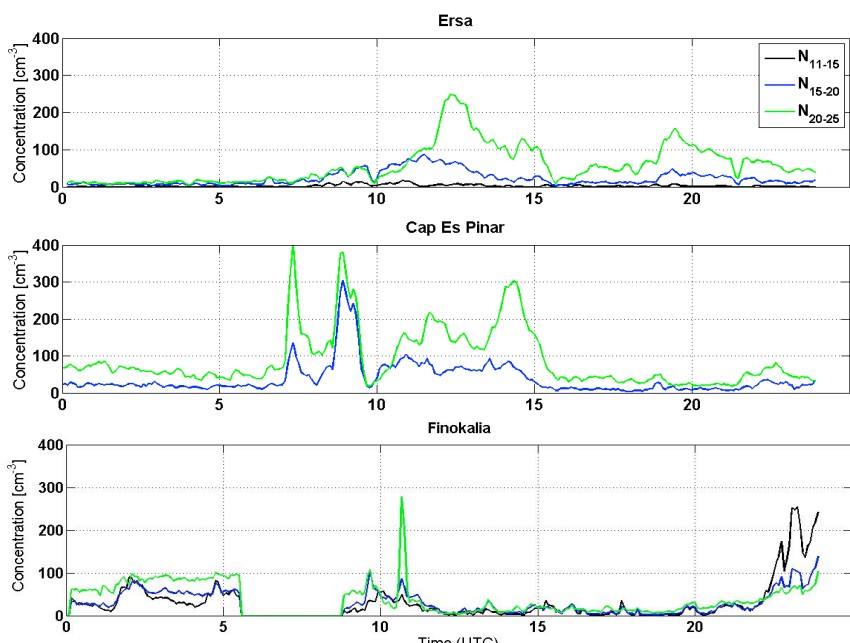

**Figure 8: Temporal evolution of the particle concentrations in the size range 11-15 nm (black) (N_{11-15}), 15-20 nm (blue) (N_{15-20}) and 20-25 nm (green) (N_{20-25}) for August 9th event.**






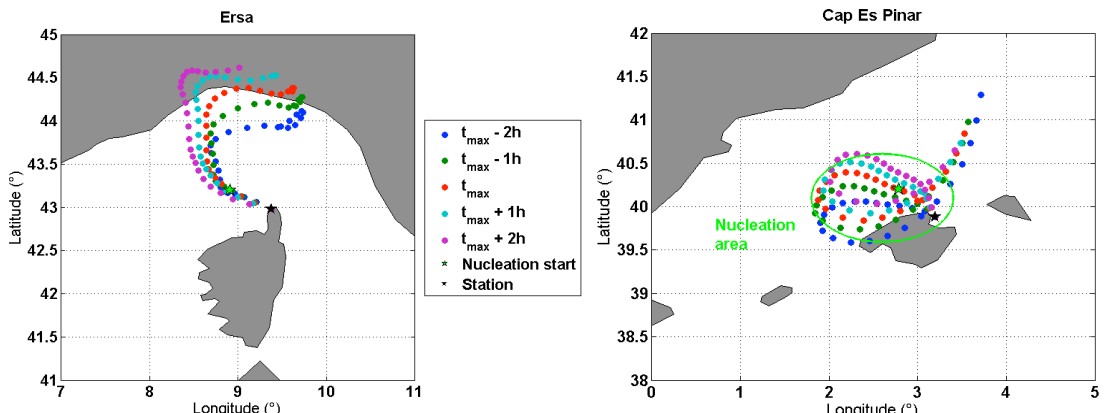

**Figure 9:** Backtrajectories of air masses sampled in Ersa and Cap Es Pinar on August 9th at tmax, when 20 nm particles concentration is maximum, and during the two hours that precede and follow this maximum. The location where nucleation initially occurs upstream the station is marked with a green star.




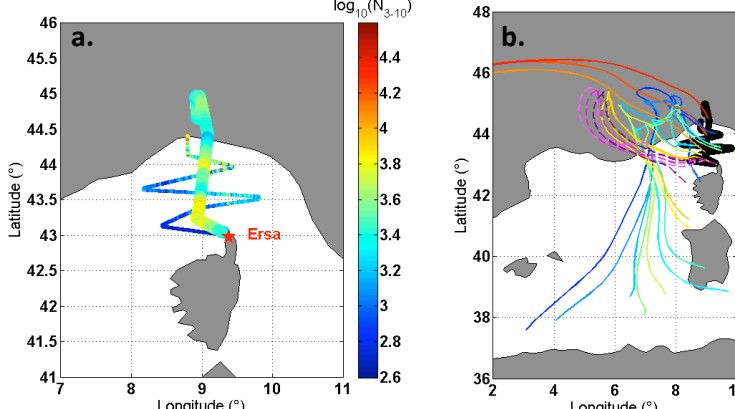


**Figure 10: a.** $N_{3-10}$ above the threshold value along the flight path performed on July 30th. Large size dots stand for
**high altitude measurements (~ 3400 m a.s.l.) while small size dots stand for low altitude measurements (~ 215 m a.s.l.);**
**b.** Air mass back trajectories calculated along the flight path (black line) every ten minutes together with the back
**trajectories of air masses arriving in Ersa each hour during the same time period (dashed lines).**









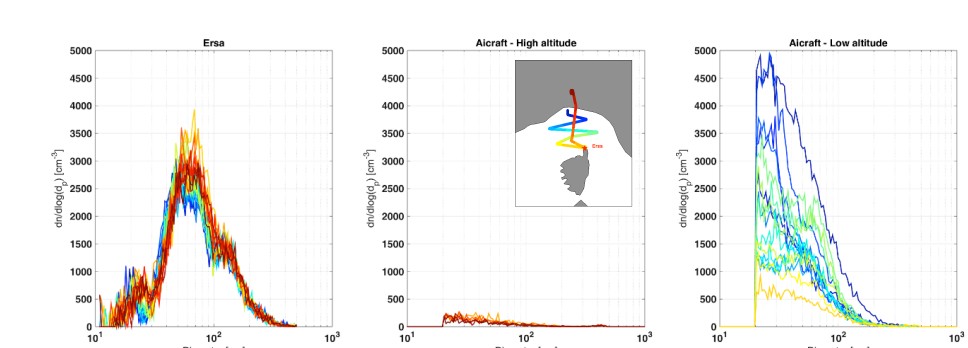

**Figure 11: SMPS size distributions measured at the Ersa station (left panel) and on board of the ATR-42 at high**
**altitudes (~ 3400 m a.s.l.) (middle panel) and low altitude (~ 215 m a.s.l.) (right panel) on July 30$^{th}$. The color coding**
**of the size distributions corresponds to the location of the aircraft, as shown on the insert of the middle panel.**



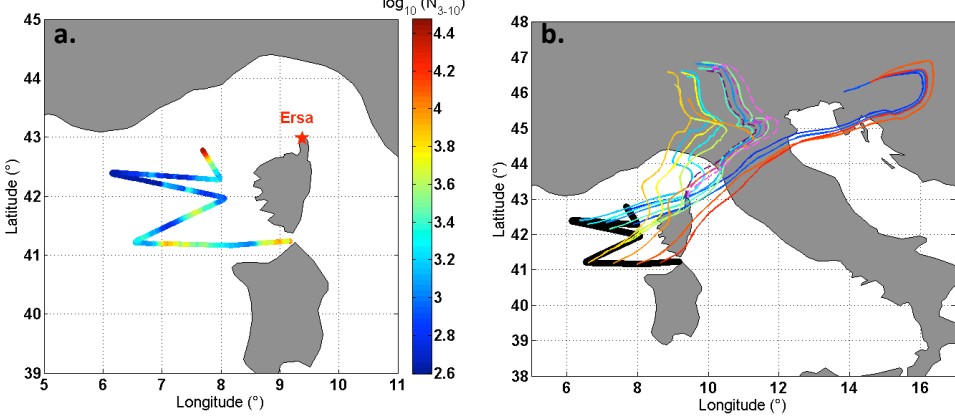


**Figure 12: a. N$_{3-10}$ above the threshold value along the flight path; b. Air mass back trajectories (solid lines) calculated**
**along the flight path (black line) every ten minutes together with the back trajectories of air masses arriving in Ersa**
**each hour during the same time period (dashed lines) during the August 1$^{st}$ flight.**







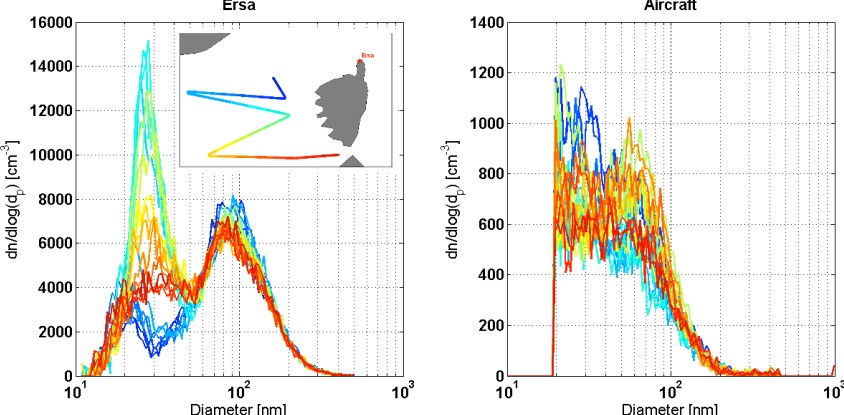


**Figure 13: Ground based (left panel) and airborne (right panel) SMPS size distributions measured on August 1$^{st}$ . The color coding of the spectra corresponds to the location of the aircraft, as shown on the insert of the left panel.**




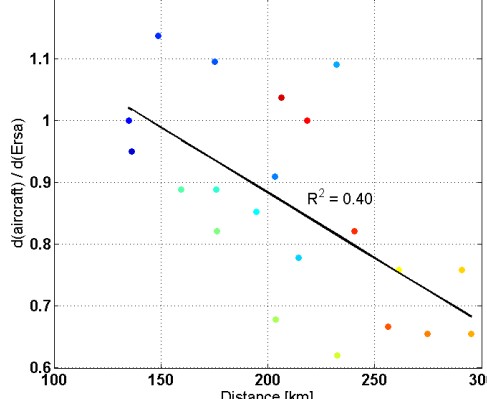


**Figure 14: Ratio of nucleation mode diameters measured onboard the ATR-42 over that calculated in Ersa as a function of the distance between the aircraft and Ersa for. August 1$^{st}$. The color coding of this scatter plot matches with the location of the aircraft showed on the insert of the left panel figure 13.**

