# Peer review of "Spatial Extent of New Particle Formation Events over the"

_Atmospheric Chemistry and Physics, 2016_

## Referee Comment (RC1) · Anonymous Referee #1 · 10 Jan 2017

Manuscript: Berland et al.: Spatial Extend of New Particle Events over the Mediterranean basin from multiple groundbased and airborne measurements.

General comments: The manuscript offers an important overview of both the horizontal and vertical spatial characteristics of new particle formation in the Mediterranean Basin (Corsica, Crete, and Mallorca) through the use of multiple ground-stations and airborne data. Additionally, it compares case studies of days presenting continental versus marine onset locations of new particle formation. The data provides an important addition to new particle formation (NPF) characteristics in the Mediterranean, by providing regional year-long comparison between the Mediterranean islands, and a differentiation in characteristics between NPF source origins (marine vs. continental) and

altitude. The measurements form part of the CHARMEX-ADRIMED and CHARMEX-SafMed projects. The manuscript is overall easy to read and follow. However, I would have liked to see meteorology data when comparing differences in NPF characteristics across the sites, in addition to the back trajectories and wind direction. The paper would have been stronger with a more complete comparison of meteorological conditions, particularly on the seasonality comparison between Ersa and Finokalia. I felt this limited the wider scope this paper could have had in identifying NPF processes in the Mediterranean, although overall, the study receives merit in providing a good, general overview of Mediterranean NPF characteristics including formation and growth rates as well as indicating both marine and continental origins of NPF. I recommend the following revisions, after which I would recommend for publication in ACP.

Specific comments (line number inside parenthesis): (Line 79) Which frequencies are 25-36%? In Finokalia, Spain, or both?

In Section 4.1.1 you provide percentage of data, but please include how many measurement days you had so that the % becomes meaningful to the readers and we can assess the strength of the statistics you are giving us. Please also include number of bad/discarded data days. This could be a table. You mention number of event days in section 4.1.2. Please transfer to section 4.1.1 and expand for each class.

(189) Add a reference to the spring annual maxima in NPF occurrence (such as reference to Maninnen et al. 2010)

(237) "Shows" would not be the right word. Although we expect high emissions and radiation in summer, you haven't included (and thus, 'shown') this data. However, you do refer to both radiation and emissions throughout the paper. It is important to include at least solar radiation data in your work, which I understand is available in both stations, or explain why you haven't. But unless there is no access to solar radiation for the days in this study, I would strongly argue for including radiation (and other meteorology parameters) in your analysis, as your arguments are dependent on

it.

(235-250): It is interesting the Cs differs the most between stations not between event and nonevent days, as in Hyytiälä, Finland, where there can be an order of maginitude difference for example. While I agree with your conclusion in terms of higher emissions needed in Finokalia to make up for a high Cs in summer, I don't see how Cs is really a determining factor in the important months of spring (March-April) between an event and nonevent, when Js and GRs are highest in both stations, but median Cs is similar during events and nonevents, and across both stations, but you still get ∼50% of the month being nonevents and ∼50% type1&2 events. Perhaps for Spring, another factor is equally or more important than Cs (which has low levels in spring). This is just my observation.

(254-255) The conclusion of deriving the number of event days to the an order of maginute less than the distance between the station seems unfounded. It is not clear how you arrived to this conclusion, other than the numbers differing by a factor or x10. Please expand explanation.

(272-273): It's not clear how/based on what you chose the specific days of 5th and 29th of July (eg. why 5th of July instead of 4th, based on Fig.7). You do mention in the next section 4.2.2. that 9th August had the most similarities in all 3 sites, although there was an instrumental breakdown at Finokalia in the morning that prevented a full interstation comparison. And you have airborne data see an event on July 30th and Aug 1st, why did you not choose a day for a horizontal (3 stations) + vertical (airborne) analysis? While the 3 days are indeed interesting, it would be good to know what we are missing or not missing from the other days. Please briefly explain your decision.

Conclusion: The first part of the analysis is the year long comparison between Finokalia and Ersa, which resulted in similar median NPF characteristics. The day case studies however, focus on Ersa and Mallorca, with more difference found in Finokalia. It may be interesting to expand the conclusions that can be made from long term single median

values and their representation of the sites and processes, compared to analysis case studies.

Figure 7: Please include colorbar for the number concentration.

Technical corrections:

(Line 1). Consider adding 'formation' in the title, as you are specifically referring to NPF.

(24). 'to analyze'.

(26). Consider substituting the word 'Globally' since it can be ambiguous ('worldwide', or 'generally').

(32) 'a daily scale'

(43). Chronological order of references. The same throughout the text.

(46). Consider editing line to: 'and use parameterizations which are based on a limited number of mechanisms. . .'

(55-56). Manninen et al. 2010 precisely does do an spatial extent analysis of NPF.

(67). rephrase to: "up to 500 km away from". . .

(76). rephrase to: "only a few studies related to NPF". . .

(83). rephrase to: "exposed to high solar radiation". . .

(86). Only French community?

(91) add word: "the long term analysis"

(97) rephrase to: "at a daily scale"

(103) remove comma in "(Mallorca,)".

(104) remove repeated year in "October 15th 2013, 2013)"

(112) rephrase to "South-Eastern Europe"

(115) 'type' is not necessary. Remove caps to make it the same style as in line 130 ('scanning mobility particle sizer)

(123) consider repharsing to "with wind speeds stronger than..."

(136) lowercase K in 'Km'.

(144) remove 'in', or remove parenthesis before author's name "'Crumeyolle et al.".

(145) remove comma in (TSI, 3010)

(156) rephrase to "NPF days", and add period in "et al."

(183) rephrase to "are very similar at Finokalia and Ersa, being 36% and 35%, respectively."

(191) rephrase to "in the different event categories"...

(195) 'Globally" can be misunderstood to mean that worldwide these events are most frequent. Subsitute word, perhaps 'annually'.

(251) please rephrase "Globally".

(275) "Table 2", instead of "Tab. 2"

(295-297): You first refer to Figure 9 before Figure 8. Please add 'see Fig. 9, since the present order of the figures does not need to be changed.

Figure fonts could be larger, particularly Fig.7 and 11, and thickness/color of lines in Fig.8.

Please also note the supplement to this comment:
http://www.atmos-chem-phys-discuss.net/acp-2016-931/acp-2016-931-RC1-supplement.pdf

---

## Referee Comment (RC2) · Anonymous Referee #2 · 3 Mar 2017

This manuscript makes an attempt to investigate the spatial extend of atmospheric new particle formation (NPF) over the Mediterranean area. The study is based on continuous measurements at 3 sites, along with air craft measurements. The topic of this study is definitely important, but the conducted analysis is not deep enough at present stage. Before this paper can be considered for publication, the authors should carefully consider and address the following issues:

I wonder why the authors chose 16 nm for calculating the particle formation rate (and minimum size for calculating GR). In both Ersa and Finokalia, size distribution measurements are available down to about 10 nm. Values of $J_{10}$ are much better comparable to other studies than $J_{16}$.

[Figure]

While equation 1 is mathematically correct, the last correction term in it is based on a very narrow size range. This can make J very sensitive to this correction term. Have the authors investigated this sensitivity? An additional problem related to this is that also GR in determined based on this very narrow size range. The authors state that the median GR in Finokalia is slightly larger than GR reported in an earlier study for a wider size range (16-20 nm vs 7-20 nm, lines 202-205). However, the difference is not slight at all, but a factor of 4! This larger difference makes me suspicious about reliability of GR determined here using the very narrow size range. This problem concerns also the GR calculated for Ersa: Figure 4 shows a few very high (= unrealistic) monthly-mean GR values.

I wonder why the authors did not report how frequently NPF takes place during the same days between the different station pairs. This kind of information is quite essential when investigating the spatial extend of atmospheric NPF.

The concept "nucleation area" should be explained better than done here in the main text. By the way, 9 km or 40 km does not represent area, but rather a diameter or some other length measure of an area.

The authors state that particle size distributions showed similar trends in Ersa and Cap Es Pinar during the intensive campaign (line 264). By simply looking at Figure 7, I cannot agree with this statement. First, the time axis of this figure is so squeezed that it is almost impossible to detect diurnal evolution of size distributions during individual days. Second, the occurrence of NPF event starting from the lowest sizes (10-20 nm) do not seem to co-inside very well between these two stations.

In addition to the couple of studies mentioned in the introduction, the authors should summarize/discuss a few other earlier studies in which the spatial extend of regional NPF has been studied using multiple stations. This could be done either in introduction, or later in the paper when discussing the results in more detail. Examples of such studies include: Vana et al 2004, JGR 109, D17201; Komppula et al 2006, Atmos

Chem Phys 6, 2811-24; Hussein et al. 2009, Atmos. Chem Phys 9, 4699-4716; Jung et al, 2013, Atmos Chem Phys 13, 51-68; Jun et al 2014, Atmos Pollution Res 5, 447-454; Kim et al 2016, Atmos Res 168, 80-91; Salma et al 2016, Atmos Chem Phys 16, 8715-28.

The main stated result of this paper is that the spatial extend of NPF is several hundreds of km over Mediterranean. I am not fully convinced that the results really show this because 1) the estimated nucleation areas are rather small (10-40 km in length), 2) it remains unclear how frequency NPF is observed in at least 2 of the stations during the same day, and 3) the available air craft data do not really support this statement either.

Minor things

Lines 114, 131 and 137: the reported size ranges should be given in proper accuracy. 2 digits would sound better than 4 digits for two of these stations.

Line 222: frequency is not weak but low.

Line 229 and later in the text: A widely used acronym for condensation sink is CS, not Cs as used here.

Line 251: globally?

Line 309: may?

---

## Author Comment (AC1) · 31 May 2017

We thank Reviewer N°1 for his comments and suggestions, which we hope will help improving the manuscript. We have addressed the comments point by point below.

Specific comment 1: (Line 79) Which frequencies are 25-36%? In Finokalia, Spain, or both?

Reply 1: These frequencies are for Finokalia, which is now clearly indicated.

SC2: In Section 4.1.1 you provide percentage of data, but please include how many measurement days you had so that the % becomes meaningful to the readers and we can assess the strength of the statistics you are giving us. Please also include number

of bad/discarded data days. This could be a table. You mention number of event days in section 4.1.2. Please transfer to section 4.1.1 and expand for each class.

Reply2: As suggested, a Table was included in Section 4.1.1, providing the total number of measurement after filtering bad data, the number of event days for each event type (I and II), the number of undefined days and the number of non-event days.

SC3: (189) Add a reference to the spring annual maxima in NPF occurrence (such as reference to Maninnen et al. 2010)

Reply 3: Reference to Maninnen et al. (2010) was transferred from l191 to l190.

SC4: (237) "Shows" would not be the right word. Although we expect high emissions and radiation in summer, you haven't included (and thus, 'shown') this data. However, you do refer to both radiation and emissions throughout the paper. It is important to include at least solar radiation data in your work, which I understand is available in both stations, or explain why you haven't. But unless there is no access to solar radiation for the days in this study, I would strongly argue for including radiation (and other meteorology parameters) in your analysis, as your arguments are dependent on it.

Reply4: A figure showing the seasonal variations of temperature (affecting emissions of biogenic precursors) and radiation (affecting the oxidation of these biogenic vapors) was added to the supplementary. Also, the discussion in the main text has been slightly developed compared to initial manuscript: "As previously suggested by Manninen et al. (2010, and references therein) and further supported by Fig. S1, higher NPF frequencies in spring are most probably related to the onset of biogenic emissions which is favored by increasing temperatures, together with higher solar radiation enhancing the production of low volatile oxidized vapors".

SC5: (235-250): It is interesting the Cs differs the most between stations not between event and nonevent days, as in Hyytiälä, Finland, where there can be an order of

maginitude difference for example. While I agree with your conclusion in terms of higher emissions needed in Finokalia to make up for a high Cs in summer, I don't see how Cs is really a determining factor in the important months of spring (March-April) between an event and nonevent, when Js and GRs are highest in both stations, but median Cs is similar during events and nonevents, and across both stations, but you still get ≈50% of the month being nonevents and≈50% type1&2 events. Perhaps for Spring, another factor is equally or more important than Cs (which has low levels in spring). This is just my observation.

Reply 5: This is actually a good remark. Additional discussion is now included in the manuscript: "One should however note that during spring months (especially March and April), median CS is similar on event and non-event days. This observation suggests that during this period, the strength of precursors emissions together with radiation might be driving the occurrence NPF to a major extent."

SC6: (254-255) The conclusion of deriving the number of event days to the an order of maginute less than the distance between the station seems unfounded. It is not clear how you arrived to this conclusion, other than the numbers differing by a factor or x10. Please expand explanation.

Reply 6: We did not aim at connecting those numbers (number of event days vs distance between the sites), which would have of course been unfounded. The purpose of the sentence was only to highlight the fact that observing events from these two stations on similar days could suggest a large spatial extent of NPF, in the order of 1000km, which is the distance between the stations. The sentence was slightly change to avoid misunderstanding.

SC7: (272-273): It's not clear how/based on what you chose the specific days of 5th and 29th of July (eg. why 5th of July instead of 4th, based on Fig.7). You do mention in the next section 4.2.2. that 9th August had the most similarities in all 3 sites, although there was an instrumental breakdown at Finokalia in the morning that prevented a full

interstation comparison. And you have airborne data see an event on July 30th and Aug 1st, why did you not choose a day for a horizontal (3 stations) + vertical (airborne) analysis? While the 3 days are indeed interesting, it would be good to know what we are missing or not missing from the other days. Please briefly explain your decision.

Reply 7: The aim of the comparison reported in Section 4.2.2 was to investigate NPF at the three stations, in terms "timing" at the day scale and "strength", especially for the closest sites (Ersa and Cap Es Pinar). This analysis thus relies on formation and growth rates calculations. The 3 specific days included in the analysis are those for which such calculations could be performed with a sufficient level of confidence (type I events) both for Ersa and Cap Es Pinar (this is now mentioned in the text), and unfortunately do not include those days for which NPF was also detected from the ATR-42: "Type one events were observed in Ersa and Cap Es Pinar on those specific days, thus allowing for particle formation and growth rates calculations, and further direct comparison of event intensity at these two sites."

SC8: Conclusion: The first part of the analysis is the yearlong comparison between Finokalia and Ersa, which resulted in similar median NPF characteristics. The day case studies however, focus on Ersa and Mallorca, with more difference found in Finokalia. It may be interesting to expand the conclusions that can be made from long term single median values and their representation of the sites and processes, compared to analysis case studies.

Reply8: We added a comment in the conclusion addressing this aspect: "The case studies also showed that despite the fact that nucleation monthly frequencies, monthly nucleation rates and growth rates had similar seasonnal variations in Ersa and Finokalia, different behaviors were observed on a daily basis between the western and eastern mediterranean bassins. Again, the combination of favourable synoptic conditions and seasonnal variations in general emission schemes may favour a seasonnal behavior of the NPF frequency and characteristics, but local conditions are modulating the general behavior of regional NPF."

SC9 Figure 7: Please include colorbar for the number concentration.

Reply 9: The colorbar already existed but Fig. 7 was too big and part of it was cropped in when editing the manuscript in ACPD. This should now be fine.

Technical corrections: they were all addressed.

———————————————

---

## Author Comment (AC2) · 31 May 2017

We thank Referee N°2 for his comments and suggestions that were very useful for improving the manuscript.

Comment 1: I wonder why the authors chose 16 nm for calculating the particle formation rate (and minimum size for calculating GR). In both Ersa and Finokalia, size distribution measurements are available down to about 10 nm. Values of J10 are much better comparable to other studies than J16.

Reply 1: It is true that providing J10 instead of J16 would have ease the comparison with other studies. However, as can be seen in Fig. 7, sub-16 nm concentrations were

most of the time very noisy in Cap Es Pinar, most probably because of a sampling line instrumental issue, and thus did not systematically allow for J10 calculation. This is now clearly stated in Section 3.2, and the fact that the comparison might be done carefully with J10 is now also explicitly mentioned, both in Section 3.2 and 4.1.2: "While formation rates (J) are usually calculated for 10 nm particles (J10), sampling line issues causing high variability of the sub-16 nm concentrations in Cap Es Pinar (see Fig. 7) only allowed for calculations involving larger diameter particle concentrations (J16). In order to ease the comparison between Ersa and Cap Es Pinar, a similar size range was applied for J calculation from the Ersa dataset. For comparison with the literature, one has to keep in mind that J16 are lower than J10, due to coagulation effects during the growth of the particles from 10 nm to 16 nm." "Besides different environmental conditions which might explain these differences, one has to keep in might that J16 values are expected to be lower than J10 because of the coagulation processes which cause particle loss during their growth."

Comment 2: While equation 1 is mathematically correct, the last correction term in it is based on a very narrow size range. This can make J very sensitive to this correction term. Have the authors investigated this sensitivity? An additional problem related to this is that also GR undetermined based on this very narrow size range. The authors state that the median GR in Finokalia is slightly larger than GR reported in an earlier study for a wider size range (16-20 nm vs 7-20 nm, lines 202-205). However, the difference is not slight at all, but a factor of 4! This larger difference makes me suspicious about reliability of GR determined here using the very narrow size range. This problem concerns also the GR calculated for Ersa: Figure 4 shows a few very high (= unrealistic) monthly-mean GR values.

Reply2: The choice of 20 nm as an upper limit for GR calculation was driven by the fact that in many cases, particle growth beyond 20 nm was not linear. We however investigated the variability of the GR using different size ranges (16-20 nm and 15-25 nm) for the three case studies discussed in the second part of the paper. Based on

this sensitivity study, it seems that the variability of the calculation within a given size range is higher than between the two size ranges. However, we cannot ensure that comparing GR16-20 with GR7-20 would lead to similar conclusions, so comparison with the literature is now performed with emphasis on the uncertainty on the GR calculation, due to both high size range and small size interval that was chosen for the calculations. " The values obtained at Finokalia are in the upper range of the values reported by Manninen et al. (2010) at European sites for 7 – 20 nm diameter particles (1.8 – 20 nm h-1, mean value 4.4 nm h-1). Especially, the values calculated in this work are on average higher compared to those obtained at other European coastal sites such as Cabauw (2.1 - 19 nm h-1, mean value 6.7 nm h-1) and Mace Head (2.7 – 10 nm h-1, mean year value 5.4 nm h-1) (Manninen et al., 2010). Higher growth rates are expected in environments with high solar radiation and emissions, such as the Mediterranean basin. However, the median value reported here is also higher than the one reported for Finokalia from the years 2008-2009 in the size range 7 – 20 nm (5 nm h-1) (Manninen et al., 2010). This result may be explained by the higher size range used here for the GR calculation (16-20nm instead of 7-20 nm), which leads to higher values because GR usually increases with particle size, but also higher uncertainty because of the narrow size range. " Also, the fact that GR are indeed high is expected for high radiation and emission areas.

Comment 3: I wonder why the authors did not report how frequently NPF takes place during the same days between the different station pairs. This kind of information is quite essential when investigating the spatial extend of atmospheric NPF.

Reply 3: The information regarding long-term measurement in Ersa and Finokalia is already provided in the text (l252-254). Concerning the intensive campaign, the information is available in Table S1. We have however included one additional sentence in Section 4.2.1:" As reported in Table S1, during this 41-days period, NPF was observed to occur at one station (at least) on 23 days. Among these 23 event days, 8 events were observed on the same day on two stations at least. This frequency of simultaneous NPF events occurrence is very similar to the one observed at Korean coastal sites (5 out of 21 observation days, Kim et al. 2016). NPF was detected at all sites on August 9th, and three events were reported on the same day for each of the station pairs Ersa – Finokalia and Ersa – Mallorca, and one event for the pair Finokalia - Mallorca."

Comment 4: The concept "nucleation area" should be explained better than done here in the main text. By the way, 9 km or 40 km does not represent area, but rather a diameter or some other length measure of an area.

Reply 4: The method we used to estimate the location where nucleation is triggered upstream the station is now explained in the main text (Section 4.2.2) rather than in the supplementary. It is true that most of the information we provide is distance instead of area, so the text was changed accordingly when necessary. Eg: "On July 5th, previous calculations lead to distances of at least 9 km (Ersa) and 40 km (Cap Es Pinar) upstream the stations, which thus cannot allow further conclusions on the simultaneity of a large NPF covering the spatial area of both stations."

Comment 5: The authors state that particle size distributions showed similar trends in Ersa and Cap Es Pinar during the intensive campaign (line 264). By simply looking at Figure 7, I cannot agree with this statement. First, the time axis of this figure is so squeezed that it is almost impossible to detect diurnal evolution of size distributions during individual days. Second, the occurrence of NPF event starting from the lowest sizes (10-20 nm) do not seem to co-inside very well between these two stations.

Reply 5: As mentioned in the title of section 4.2.1, the aim of Fig. 7 is only to provide a global overview of the time evolution of the particle size distribution at the three stations during the intensive campaign. We clearly believe that at this "campaign scale", Fig. 7 highlights 3 sub-periods during which all three stations display higher nucleation frequencies. However, we agree with the fact the comparison between the sites cannot only rely on this global approach, that is why Section 4.2.2 is dedicated to a more detailed analysis to describe the similarities/differences between the events observed

on the same days at the three stations.

Comment 6: In addition to the couple of studies mentioned in the introduction, the authors should summarize/discuss a few other earlier studies in which the spatial extend of regional NPF has been studied using multiple stations. This could be done either in introduction, or later in the paper when discussing the results in more detail. Examples of such studies include: Vana et al 2004, JGR 109, D17201; Komppula et al 2006, Atmos Chem Phys 6, 2811-24; Hussein et al. 2009, Atmos. Chem Phys 9, 4699-4716; Jung et al, 2013, Atmos Chem Phys 13, 51-68; Jun et al 2014, Atmos Pollution Res 5, 447454; Kim et al 2016, Atmos Res 168, 80-91; Salma et al 2016, Atmos Chem Phys 16, 8715-28.

Reply 6: We thank the reviewer for this useful list of references. We used the references for works related to comparisons of NPF events detected at multiple background sites, but the ones involving urban areas, which are very specific and would not help understanding our results.

Comment 7: The main stated result of this paper is that the spatial extend of NPF is several hundreds of km over Mediterranean. I am not fully convinced that the results really show this because 1) the estimated nucleation areas are rather small (10-40 km in length), 2) it remains unclear how frequency NPF is observed in at least 2 of the stations during the same day, and 3) the available air craft data do not really support this statement either.

Reply 7: 1) One of the methodologies used in this paper to assess the spatial extend of NPF in the Mediteranean area, (i.e. investigating similarities in NPF time occurrence between several stations) is very similar to the one used by several authors that draw the same conclusion for other environments. We additionally calculated the minimum areas in which nucleation occurred. The fact that our calculation gives a minimum area, and not the totality of the nucleation spatial extend is now better explained in the text. 2) This information was present in the manuscript, but it is now better highlighted in

the conclusion: "NPF formation was observed to occur simultaneously at least at two of the three stations on 8 days over the 41 days of observation, which confirms the frequent occurrence of regional scale NPF events in the Mediterranean area. " 3) Aircraft data do show that NPF occurs over a large spatial area, but give additional information on geographical gradients and hence indicate that the regional NPF event may have different sources (continental, marine, high altitude). This is now better specified in the conclusion: "Airborne measurements confirmed the regional spatial extend of NPF events, and further showed regional NPF events can have different sources. The selected events depicted contrasting situations where particles were initially probably formed above the continent for one of them, both in the boundary layer and in the free troposphere, and probably formed above the sea for the other."

Minor comments: they were all addressed

———————————————————